A systematic review on the morphology structure, propagation characteristics, resistance physiology and exploitation and utilization of Nitraria tangutorum Bobrov.

Li Xiaolan
Liu Hanghang
Li Chaoqun
Li Yi liyi@gsau.edu.cn
Gansu Agricultural University , Lanzhou , China
Nikalje Ganesh
Electronic publication date: 2024 Aug 16
Publication date: 2024
Volume: 12
Electronic Location ID: e17830
Received 2024 Mar 12; Accepted 2024 Jul 8
Copyright: ©2024 Li et al.
Copyright year: 2024
Copyright holder: Li et al.
License: This is an open access article distributed under the terms of the Creative Commons Attribution License, which permits unrestricted use, distribution, reproduction and adaptation in any medium and for any purpose provided that it is properly attributed. For attribution, the original author(s), title, publication source (PeerJ) and either DOI or URL of the article must be cited.
License URL: https://creativecommons.org/licenses/by/4.0/

Keywords: Nitraria tangutorum, Morphology structure, Propagation characteristics, Resistance physiology, Exploitation and utilization

Funding: China’s Science and Technology Support Poject for Developing Countries KY202002011 Central Government’s Extension and Demonstration Fund for Forestry Science and Technology [2020]ZYTG15 Forestry Science and Technology Innovation and International Cooperation of Gansu Province GLC2019-418-8 Gansu Provincial Science and Technology Program No. 24JRRA677 This study was supported by the China’s Science and Technology Support Project for Developing Countries (KY202002011); the Central Government’s Extension and Demonstration Fund for Forestry Science and Technology ([2020]ZYTG15); and the Fund for Forestry Science and Technology Innovation and International Cooperation of Gansu Province (GLC2019-418-8). Gansu Provincial Science and Technology Program (No. 24JRRA677) funded the APC of the article. The funders had no role in study design, data collection and analysis, decision to publish, or preparation of the manuscript.

==============================
Nitraria tangutorum Bobrov., belonging to the family Nitrariaceae, is a drought-tolerant and salt-loving plant and has drawn attention for its good economic and ecological value. As one of the main group species and dominant species in China’s desert and semi-desert regions, N. tangutorum possesses superior tolerance to drought, high temperature, cold, barren, high salinity and alkalinity and wind and sand. Its root system is well developed, with many branches and a strong germination capacity. Once buried in sandy soil, N. tangutorum can quickly produce a large number of adventitious roots, forming new plants and continuously expanding the shrubs, forming fixed and semi-fixed shrub sand dunes. Sand dune shrubs can trap and fix a large amounts of quicksand, prevent desert expansion and erosion, and play an important role in maintaining regional ecosystem balance and improving ecological environmental quality. In addition, the phytochemical screening studies report that N. tangutorum contains an abundance of various compounds including flavonoids, alkaloids, phenolic acids and polysaccharides. These compounds confer a range of beneficial bioactivities such as antioxidant, anti-inflammatory, anti-tumor, anti-fatigue, liver protection, neuroprotection, cardiovascular protection, lowering blood lipid, regulating blood sugar level and immunoregulation. The fruits of N. tangutorum also contain vitamin C, amino acids, minerals and microelements. It has been traditionally used as a nutritional food source and in folk medicine to treat diseases of the spleen and stomach, abnormal menstruation, indigestion, and hyperlipidemia. N. tangutorum, as a wild plant with medicinal and edible homology, possesses remarkable economic and medicinal values. This detailed, comprehensive review gathers and presents all the information related to the morphological structure, propagation characteristics, resistance physiology and exploitation and utilization of N. tangutorum, providing a theoretical basis for the researchers to conduct future in-depth research on N. tangutorum.

Introduction

Nitraria tangutorum Bobrov. is a perennial deciduous shrub that belongs to the Nitratia genus in the Nitrariaceae family. This plant is native to China and is an endemic species to our country (Li, 2011) (Fig. 1). It is mainly distributed in the arid desert, alpine region and saline-alkali areas of northwest China. It exhibits high tolerance to drought, cold, sand burial and salinity (Zhang et al., 2017). This typical plant has developed roots and numerous branches that can effectively fix quicksand and reduce wind speed, making it a pioneer tree for wind prevention and sand fixation in desert areas (Abla et al., 2019). N. tangutorum is a wild plant with medicinal and edible homology, which has high economic, ecological and medicinal value (Yang et al., 2010). The fruit of N. tangutorum is a rare wild berry in the desert that tastes sweet and sour and is called “desert cherry”. This plant contains a variety of bioactive compounds, including alkaloids, flavones, vitamins, amino acids, and polysaccharides. It has been traditionally used in folk medicine to treat diseases of the spleen and stomach, abnormal menstruation, and hyperlipidemia due to its antioxidant properties (Yang, Suo & Wang, 2012). The branches and leaves of N. tangutorum are rich in amino acids, linoleic acid, crude protein, crude fat, as well as some important mineral elements such as phosphorus, iron, calcium, and zinc, etc. which have high nutritional value. These nutrients make the branches and leaves of N. tangutorum a good feed for the development of animal husbandry (Wu et al., 2017). The oil content of N. tangutorum seed is 11% to 13%, which is a rare functional oil rich in unsaturated fatty acids in nature. The seed oil of N. tangutorum has been shown to have obvious health effects in protecting against liver injury (Suo, Wang & Chen, 2005), lowering blood lipid (Suo, Gao & Wang, 2005), and anti-fatigue (Suo et al., 2006). Moreover, Cynomorium songaricum Rupr., a valuable herb in traditional Chinese medicine, primarily parasitizes on the roots of N. tangutorum. It has an ideal effect on enhancing immunity, anti-aging, endocrine regulation and treating senile diseases and is known as “desert ginseng”. Therefore, N. tangutorum has both ecological value for sand prevention and control and economic value for food and medicine, making it a promising plant resource with great potential for development and utilization.

Figure 1 Provincial distribution map of N. tangutorum in China.

Map source credit: Runfola et al. (2020) (https://www.geoboundaries.org/).

SURVEY METHODOLOGY

PubMed, Web of Science, Wanfang, Google Scholar, Research Gate, and the China National Knowledge Infrastructure databases were searched for relevant articles. 318 articles appeared in the database using “Nitraria tangutorum”, “morphology structure”, “propagation characteristics”, “resistance physiology”, and “exploitation and utilization” as the search term, and the date of publication was from 1982/8/29 to 2024/2/2. After removing duplicate articles and articles with little relevance, 137 articles were selected for review.

Figure 2 The morphological characteristics of N. tangutorum.

(A), (B), (C) and (D) are from the Gansu Province Academy of Qilian Water Resource Conservation Forests Research Institute. Photo credit: Xiaolan Li.

MORPHOLOGY STRUCTURE

Plants that have evolved in a particular environment over a long period of time exhibit relatively consistent adaptive traits, such as morphological and structural features (Guo, 2018). N. tangutorum is a typical xerophytic diving shrub that grows in desert and semi-desert lake basin sandy land, river terraces, and clayed ground with accumulated aeolian sand. The plant is 1–2 m tall, multiple branches, prostrate and apex needle-like. The branches buried by sand are easy to produce adventitious roots, forming shrub sand dunes (Fig. 2A), which are beneficial for fixing quicksand (Zuo, 2013). The leaves are usually 2–3 fascicled, broadly oblanceolate, 18–25 mm long, 6–8 mm wide, apex obtuse-rounded or flat-truncated, base tapering into a cuneate, with entire margins (Fig. 2B). Plant leaves are the most sensitive organs to environmental changes, and their structural characteristics best reflect the adaptability of plants to the environment (Li, 1981). Zuo’s (2013) research showed that the leaves of N. tangutorum have developed palisade tissue and degraded spongy tissue, and its water storage tissue has developed. Under conditions of drought or high salt concentration, mucocytic cells also appear in the mesophyll tissue, increasing the osmotic potential of the cells and facilitating water absorption (Zhang et al., 2021a; Feng, 2020). The flowers of N. tangutorum are densely arranged, white, petals and ovary glabrous. The drupe is oval, 8–12 mm long and 6–9 mm wide, deep red in color when ripe and a rose colored juice (Fig. 2C). The seed is narrow ovate, 5–6 mm long and 3–4 mm wide, apex short acuminate and significant differences in size (Zhang et al., 2005). The morphological characteristics of pollen and seeds are relatively conservative and less affected by the external environment, which is important for the identification of plant species and varieties (Fig. 2D). Zhang et al. (2019) showed in their experiments on seed morphology and germination characteristics that small seeds germinate the earliest and have a greater advantage in the early stage of germination, while large seeds germinate later and have the highest germination rate. Li et al. (2023a) measured and analyzed pollen of six families from different geography N. tangutorum by scanning electron microscopy. They found that pollen morphology of N. tangutorum was relatively consistent, medium size pollen grains, long sphere shape, three-pore groove and the germination groove was wide. The ridge of the germinal groove is small at two poles, large in the middle and increasing in a “convex” shape. The surface textures are striped and brain-shaped, and do not form a pattern. Plant buds play a crucial role in the life history of woody plants. Different types of buds consume different nutrients and perform different functions. Li et al. (2023b) surveyed the association between the allometry of N. tangutorum aboveground bud (dormant buds, vegetative buds and reproductive buds) and plant carbon (C), nitrogen (N), and phosphorus (P) contents and ratios. The result showed that allometry of three bud pattern traits shows a positive correlation with plant tissue P content, and the C:P and N:P ratios. The root system of N. tangutorum is well-developed and the tip of the lateral root occasionally forms sand sheaths, providing good effect of water retention and drought resistance (Zhang et al., 2021b).

PROPAGATION CHARACTERISTICS

The propagation methods of N. tangutorum include natural dispersal, mainly by cloning, and artificial cultivation, mainly by cutting, seed and tissue culture (Li, 2009). Clonal propagation is based on the stress development of dormant buds after their branches are buried in sand, which is the main reproductive mode of N. tangutorum (Wang, 2013). The intraspecific and interspecific hybridization of N. tangutorum was chaotic, and the seed propagation is prone to interspecific variation and intergenerational degradation. In vitro rapid propagation technique plays an important role in selective breeding of excellent varieties and resource development of N. tangutorum (Wang, 2010). Cutting propagation can maximize the preservation of the excellent traits of the parents and is the main method of artificial propagation of N. tangutorum. The study by Li Wei using combinations of different hormones (ABT, NAA, GA, IAA, IBA) and concentrations (100, 250, 500, 750 mg L−1) to treat the softwood cutting seedlings showed that the treatment with ABT 250 mg L−1 had the highest survival rate and the best treatment effect (Li, 2009). Hardwood cutting is more widely used in practical production than softwood cutting because it is simpler and easier to master. Cheng et al. (2015a) studied the effect of five hormone treatments (ABT, NAA, IAA, IBA, NAA+IBA) at different concentrations (100, 200, 400, 600 mg L−1) on the survival rate of N. tangutorum cuttings. They found that all hormone treatments significantly improved the survival rate of N. tangutorum cuttings compared to the control. Cheng et al. (2015b) also demonstrated that the effect of five hormones on survival rate was ordered as NAA > IBA > NAA + IBA > ABT > IAA > CK based on a comprehensive analysis and concluded that IBA (100 mg L−1) was the optimal treatment for N. tangutorum hardwood cutting. Tissue culture is one of the most important means to maintain the stability of the excellent properties of N. tangutorum. Zhao et al. (2014) experimented with 2–3 years old dormant branches of N. tangutorum to investigate the effects of different concentrations of NAA, IBA, ABT-1, and different treatment durations on the survival rate of cuttings. The results showed that the optimal combination for promoting the survival of N. tangutorum hardwood cutting was 50 mg L−1 ABT-1 treatment for 6 h. Guo, Lin & Wu (2009) found that tender stem with bud and leaf were excellent explant for inducing cluster bud and callus, respectively, and suitable medium for callus formation, proliferation culture and rooting culture were MS+2,4-D (1.0 mg L−1), MS+6-BA (2.0 mg L−1)+NAA (1.0 mg L−1) and 1/2MS+KT (1.0 mg L−1)+IBA (0.5 mg L−1) by screening the different explants of N. tangutorum and the appropriate culture medium. Open tissue culture is a simple and easy method compared with traditional tissue culture, and screening of antimicrobial species and concentrations is the key link in open tissue culture. Bian et al. (2022) showed in their investigation of the effects of different concentrations of sodium hypochlorite on open tissue culture seedlings of N. tangutorum that 15 to 20 mg L−1 sodium hypochlorite can be used as an antimicrobial for open tissue culture.

RESISTANCE PHYSIOLOGY

Drought resistance

Response of N. tangutorum to drought stress

Drought stress disrupts multiple physiological and biochemical processes such as nutrient absorption, photosynthesis, and cell metabolism, which severely limiting seed germination and plant growth and development (Zhang et al., 2020). Luo et al. (2014) used PEG-6000 to simulate drought conditions and investigated seed germination and seedling growth characteristics under drought stress. The results showed that both seed germination and seedling growth of N. tangutorum showed inhibitory response characteristics to drought stress. The study of leaf traits is critical to understanding how plants adapt to their habitats. Wei et al (2022) studied the effects of drought stress on N. angutorum and showed that drought stress caused a decrease in stomatal conductance, transpiration rate, intercellular CO2 concentration, and net photosynthetic rate. Zhao et al. (2020) took N. tangutorum from nine habitats in three regions as research objects and compared the morphological and structural characteristics of N. tangutorum leaves from different regions by paraffin method. The findings indicate that with the intensification of habitat drought, leaf length decreases, leaf area contracts, and leaf thickness diminishes. From a tissue structure standpoint, in drier environments, the development of palisade tissue and the number of mucilage cells increase, improving the plant’s ability to adapt to drought stress. This is consistent with the finding that intensification of environmental drought can cause thickening of palisade tissue, thinning of spongy tissue and increasing the ratio of palisade tissue and spongy tissue of N. tangutorum leaves showed in Guo (2022). Wang et al. (2023b) investigated the adaptation of different morphologies of N. tangutorum to arid environments and showed that the plants of dwarf morphology were better adapted to the drought environments. As one of the most important osmotic regulation substances, proline can remove free radicals and improve the protective effect of antioxidant enzymes in plants (Xing et al., 2018). Zhou et al. (2011) analyzed the proline concentration in N. tangutorum across three distinct precipitation regions. They discovered that N. tangutorum growing in high precipitation regions exhibited lower proline concentrations, while those in low precipitation areas displayed higher concentrations. This suggests that N. tangutorum can adapt to environmental changes by modulating its proline concentration. Wei, Li & Su (2022) treated N. tangutorum under natural drought stress with different concentrations of exogenous proline to explore the drought resistance mechanism of N. tangutorum. The results showed that N. tangutorum after treatment with different concentrations of exogenous proline alleviated the damage caused by drought through reducing the length, width and area of stomata and increasing the density of stomata.

Table 1 The tolerance level and economic value of N. tangutorum and other desert plants.

Plant name	Tolerance level	Economic value	
	Drought
(PEG-6000 MPa)	Ref.	Salt
(NaCl mM)	Ref.		
Nitratia tangutorum	−0.5	Shi et al. (2014)	400	Luo et al. (2014)	Edible, Medicinal, Forage, Ornamental plants	
Reaumuria songarica	−2.1	Liu et al. (2019b)	300	Yang & Wang (2012)	Forage, Industrial raw material	
Elaeagnus angustifolia	−1.2	Zeng et al. (2015)
Lin et al. (2015)	300	Zhang et al. (2006)	Edible, Medicinal	
Alhagi camelorum	−0.7	Li & Zhang (2013)	300	Zhao & Han (2012)	Edible, Medicinal, Forage	
Apocynum venetum	−0.7	Xiao & Wang (2020)
Fong, Yan & Liu (2018)	250	Han et al. (2021)	Medicinal, Industrial raw material	
Caragana korshinskii	−1.2	Yang et al. (2024)	200	Zhang et al. (2006)	Forage, Industrial raw material	
Lycium ruthenicum	−0.5	Zong et al. (2015)	200	Zong et al. (2015)	Edible, Medicinal	
Medicago sativa	−0.7	Zhang (2021)	200	Wang et al. (2024)	Edible, Medicinal, Forage	
Melilotus suaveolens	−0.2	Zhang (2021)	200	Zhang et al. (2024)	Medicinal, Forage, Industrial raw material	
Hedysarum scoparium	−0.7	Zou et al. (2022)	100	Zou et al. (2022)	Edible, Forage, Ornamental plants	
Artemisia desertorum	−0.3	Chen et al. (2021b)	100	Chen et al. (2021b)	Edible, Medicinal, Forage	

Drought resistance in plants is a complex and comprehensive trait that is influenced by multiple factors (Table 1). Chong et al. (2011) selected 4 populations of N. tangutorum from different geographical areas to determine and analyze 17 physiological and biochemical indices related to drought resistance. The results indicated that N. tangutorum populations in Jinta County, Jiuquan City had the strongest drought resistance. There are also differences in drought resistance among different pedigrees of the same plant, and pedigree selection is one of the important means of forest genetic improvement. Chai et al. (2017) conducted drought resistance screening on 31 N. tangutorum pedigrees from two experimental sites in Lanzhou and Wuwei, and obtained 5 drought-resistant pedigrees. After repeated verification of the experimental results by Li et al. (2020) from the same research group in different years, it was found that the N. tangutorum pedigree has stable drought resistance in different physiological stages and environmental conditions. When plants suffer from drought stress, reactive oxygen will accumulate excessively in the body, damaging the cell membrane structure and causing oxidative damage to plants (Gill & Tuteja, 2010). Anthocyanins, as a typical representative of water-soluble pigments in plants, have prominent antioxidant functions and play an important role in the environmental adaptation of plants (Tanaka, Sasaki & Ohmiya, 2008). Gao et al. (2020) cloned a key gene of anthocyanidin synthesis, the flavonoid 3-O-glucosyltransferase gene (UFGT), from the cDNA of N. tangutorum and performed functional validation. The results confirmed that NtUFGT can facilitate the activity of the plant antioxidant system by effectively promoting the accumulation of anthocyanidins, thus enhancing the tolerance of plants to drought stress. Transcription factors (TFs) participate in plant responses to biotic and abiotic stresses by specifically binding to stress-related cis-acting elements to regulate the expression of stress-responsive genes (Liang et al., 2019). Wang et al. (2023a) selected members of the BRI1-EMS-suppressor 1 (BES1) transcription factor family associated with brassolidin (BR) regulation based on full-length transcriptome data of N. tangutorum under drought stress, and showed that NtBES1-4 played an important role in the drought stress response of N. tangutorum by conducting bioinformatics analysis and expression pattern analysis under different concentrations of PEG (10% and 30%). Li et al. (2021) cloned CBL1, a member of the calcineurin B-like proteins family, from N. tangutorum and conducted expression analysis under different stress conditions. The results showed that the NtCBL1 gene may be involved in the drought stress response of N. tangutorum. Yi (2021) used N. tangutorum as material to identify 14-3-3 and BES1 gene families related to plant hormone signal transduction and to verify their drought resistance functions. qRT-PCR results showed that the expression of most 14-3-3 and BES1 family members was significantly induced by PEG and was tissue specific.

Salt tolerance

Effects of salt stress on N. tangutorum

Salt stress is a kind of universal abiotic stress that affects the growth and development of plants (Lokhande et al., 2013). It can change various physiological and biochemical processes of plants by inhibiting photosynthesis and cell division and expansion (Van, Zhang & Testerink, 2020). Zhao et al. (2023) investigated the effects of different concentrations of NaCl on the growth condition of N. tangutorum seedlings and found that NaCl stress treatment changed the morphology, structure, and physiological function of N. tangutorum. And the concentrations of NaCl below 1.6% promote its growth, and its growth begins to be inhibited and even dies as the concentrations of NaCl increase. This is consistent with the findings of Zhang et al. (2021a) who believed that a low concentration of salt stress (200 mmol L−1) could promote growth and a higher concentration of salt stress (≥300 mmol L−1) could damage the structure of N. tangutorum. Salt stress makes plants to suffer from osmotic stress firstly, causing ion imbalance and ion toxicity, damaging the chloroplast pigment system of plants and inhibits photosynthesis (Liu et al., 2019a). Yang et al. (2017) found that the content of all photosynthetic pigments in the leaves decreased slightly by measuring photosynthetic pigments of N. tangutorum leaves that being treated with 8% NaCl for 40 days. Osmotic regulation is the major physiological mechanism of plants to adapt to salt stress (Zhang & Shi, 2013; Cheng et al., 2015a). Wang et al. (2012) used N. tangutorum seedlings as material to study the changes in antioxidant enzyme activity and osmotic regulation substance content in leaves treated with different NaCl concentrations for different times. The results showed that the activities of SOD, CAT, and POD in leaves were enhanced under low concentration of NaCl (25∼100 mmol L−1) and decreased under high concentration of NaCl (200∼400 mmol L−1). The content of osmotic regulation substance Pro continues to rise with the increase of salt concentration. This conclusion is basically consistent with the research results of Fan, Yang & Cheng, (2009), who analyzed the cell growth, content of osmotic regulating substance and antioxidant enzyme activity under different concentrations of NaCl and showed that low salt concentrations (<100 mmol L−1) promoted the growth and high salt concentrations (>200 mmol L−1) inhibited the growth of callus. It was concluded that N. tangutorum could resist the damage of adverse environment by increasing the activity of antioxidant enzymes and the content of osmotic regulation substances. Previous studies on N. tangutorum have focused on the physiological characteristics of salt tolerance in field plants, potted seedlings, tissue culture seedlings, and callus. However, there are few reports about the salt tolerance of N. tangutorum suspension cells. Ni et al. (2015) used suspension cells of N. tangutorum as experimental materials to analyze the changes in cell growth status and physiological and biochemical indicators under different salt concentrations (0, 100, 150, 200, 250 mmol L−1). The results showed that salt stress had a significant effect on the growth status of N. tangutorum suspension cells, with the fastest growth rate observed under the salt concentration of 100 mmol L−1.

Salt-tolerance mechanism of N. tangutorum

Plants will adjust themselves at the molecular, cellular, and physiological levels to adapt to the external environment when they are in adversity (Rodziewicz et al., 2013) (Fig. 3). The accumulation of Na+ in plant cells under salt stress leads to the weakening of intracellular metabolic activity and the disturbance of ion balance, which limits plant growth and development (Anschütz, Becker & Shabala, 2014). Plants use Na+/H+ antiporter located in the plasma membrane or tonoplast to make efflux the excess Na+ from the cytoplasm or regionalize it in the vacuole to reduce the Na+ content in the cytoplasm. Tang et al. (2014) used the tender leaves of N. tangutorum as experimental material to clone the Na+/H+ antiporter gene (NHX) located in the tonoplast and perform expression analysis of NtNHX1. The results showed that the expression of NtNHX1 was tissue specific and induced and regulated by salt stress. Zheng et al. (2013) cloned the Na+/H+ antiporter located in the plasma membrane (NHA or SOS1) from N. tangutorum using RT-PCR and RACE techniques, and the expression analysis of NtSOS1 under different stress conditions found that NtSOS1 expression was induced by salt, high temperature, cold, and drought stress. Ca2+ plays an important role in various regulatory mechanisms of plant response to environmental stress. It is not only a key substance in the signal transduction system, but also an essential nutrient element for plant growth and development (Jiang et al., 2014). A large number of studies have shown that a certain concentration of exogenous Ca2+ can increase the concentration of free Ca2+ in the plant cytoplasm, protect the plasma membrane structure, inhibit the generation of reactive oxygen, increase the accumulation of osmotic regulation substance, and improve the stress resistance of plants. Yan et al. (2016) explored the effects of different concentrations of exogenous Ca2+ (0, 5, 10, 15, 20 mmol L−1) on photosynthesis of N. tangutorum under different concentrations of NaCl stress (100, 200, 300, 400 mmol L−1). The results showed that when the salt concentration was not higher than 300 mmol L−1, a certain concentration of exogenous Ca2+ (≤15 mmol L−1) could effectively regulate the photoinhibition caused by salt stress and improve the photosynthetic efficiency. And the regulatory effect of Ca2+ was not obvious when the salt concentration exceeded 300 mmol L−1. This is consistent with the conclusion of the research of Yuan et al., who believed that a certain concentration of Ca2+ (≤15 mmol L−1) can effectively alleviate the damage caused by salt stress (NaCl ≤ 300 mmol L−1) on N. tangutorum, and the alleviating effect of exogenous Ca2+ is not evident under high salt stress (Yuan et al., 2014). It may show the inhibitory effect.

Figure 3 Salt tolerance mechanism of N. tangutorum..

Figure source credit: Feng (2020). Changes of ultrastructure and salt-tolerant metabolism of Nitraria leaves under the salt stress. Jinzhong: Shanxi Agricultural University. https://image.medpeer.cn. Image credit: Xiaolan Li.

As the final product of gene transcription and protein modification, metabolites of plant reflect the metabolic level of organisms under different physiological and ecological conditions. Metabolomics can truly characterize the salt tolerance response of plants through the upregulation and downregulation of metabolites as well as the metabolic pathways in which the metabolites are actually involved (Saito & Matsuda, 2010). Yan et al. (2021) adopted GC-TOF-MS metabolomics to study the response mechanism of N. tangutorum to salt stress. The results showed that under 300 mmol L−1 NaCl, 11 differential metabolites dominated by organic acids regulated 6 metabolic pathways dominated by sulfur metabolism in response to salt stress. In recent years, transcriptome sequencing technology has been increasingly used to explore the transcriptional regulatory mechanisms of plant response to stress. The transcriptome can reflect the number and expression of differential genes under different stresses. Zhu et al. (2021) used PacBio SMRT sequencing technology to obtain the full-length transcriptome of N. tangutorum and showed that NHX expression is induced under salt stress by identifying and analyzing the Na+/H+ antiporter. Wang (2023) conducted transcriptome sequencing analysis of N. tangutorum leaves under salt stress and found that the NtCER7 gene, which is involved in wax biosynthesis and transport, plays an important role in the response of N. tangutorum to salt stress. The gene cloning and genetic transformation of NtCER7 showed that transgenic Arabidopsis thaliana of NtCER7 showed good tolerance to salt stress. Li et al. (2021) performed homologous gene alignment on existing transcriptome data, cloned and explored the expression patterns of N. tangutorum CBL1 and CBL2 genes under different stress conditions. The results showed that the NtCBL1 gene was significantly induced by 500 mmol L−1 NaCl and 300 mmol L−1 mannitol, and the NtCBL2 gene was significantly upregulated at 4 °C. The calcineurin B-like proteins (CBLs) and protein kinases (CIPKs) participates in the Ca2+ signaling pathway and play an important role in plant response to adversity (Kim et al., 2000; Shi et al., 1999). Lu et al. (2020) cloned the NtCIPK 9 gene from N. tangutorum and overexpressed it in Arabidopsis, indicating that transgenic Arabidopsis has higher salt tolerance. Zheng et al. (2014) isolated the NtCIPK2 gene from N. tangutorum and performed sequence analysis and expression analysis. The results showed that NtCIPK2 expression was significantly induced by salt, drought, high temperature, and cold.

EXPLOITATION AND UTILIZATION

Chemical composition

Clarifying the chemical composition of plants is a prerequisite and foundation for their development and utilization. N. tangutorum is a rare berry plant in psammophytes, which has high economic and pharmacological value (Ji, 1989; Yuan et al., 2018). In 1989, Jia, Zhu & Wang (1989) first studied the chemical composition of N. tangutorum and isolated 6 flavonoids from the seeds of N. tangutorum. Flavonoids, a group of natural substances with variable phenolic structures, have various biological activities such as anti-inflammatory, anti-oxidative, anti-tumor, cardiovascular and neuroprotective. Flavonoids are one of the main components of “medicine food homology” substances (Panche, Diwan & Chandra, 2016). Wang, Li & Suo (2004) determined by single factor and orthogonal experiments that the optimal extracting conditions of flavonoids from N. tangutorum seeds were as follows: 80 °C, solid–liquid ratio 1:6 (W/V), reflux extraction of 70% ethanol for 3 times, each time for 1.5 h. Wu et al. (2014) isolated seven flavonoids from the leaves of N. tangutorum. Gao (2014) identified 14 flavonoids from N. tangutorum fruits, seeds, leaves, and stems by high-performance liquid chromatography tandem mass spectrometry (MSn) method and compared the contents of each tissue. The results showed that the flavonoid content in the leaves of N. tangutorum was the highest. Anthocyanins, a type of water-soluble flavonoid compounds, are the main components of N. tangutorum (Ma et al., 2016). Zheng et al. (2011) and Chen et al. (2021a) identified nine anthocyanin components from the fruits of N. tangutorum, respectively. Zhang et al. (2017) identified 16 anthocyanins from by-products of N. tangutorum juice using HPLC-ESI-MS technology. Wu et al. (2013) and Wang et al. (2012) also isolated multiple phenolic acids and alkaloids in addition to flavonoid substances in the chemical composition analysis of N. tangutorum fruits. Phenolic acids are natural antioxidants that have the functions of preventing chronic diseases, anti-cancer and inhibiting neurodegeneration (Arts Ilja & Hollman Peter, 2005). Zhao et al. (2017) and Jiang et al. (2023) isolated 5 and 10 phenolic acid compounds from the fresh juice of N. tangutorum, respectively. Jia et al. (2020) also detected a variety of phenolic acids in different berry plants including N. tangutorum, which is distributed in the Qinghai-Tibetan Plateau. Plant alkaloids are important sources of traditional and modern medicines with rich biological activities (Yang et al., 2013a). Jiang et al. (2021) isolated and identified eight alkaloids from N. tangutorum fruits using UPLC-Q-TOF-MS/MS technology. Polysaccharides from natural sources have anti-inflammatory effects (Peng et al., 2014). Song (2016) and Ma (2019) used different methods to extract polysaccharide components from N. tangutorum fruits. In addition to flavonoids, phenolic acids, alkaloids, and polysaccharides, a large number of studies have shown that N. tangutorum is also rich in essential amino acids (Suo, Wang & Chen, 2005), trace elements (Wang et al., 2014), and vitamin C (Zhao et al., 1994).

Pharmacological value

N. tangutorum is rich in various nutrients and active ingredients with a wide range of pharmacological effects. Ma Hui’s study on the chemical constituents and activities of N. tangutorum showed that the fruits of N. tangutorum have good anti-oxidative, anti-tumor, and anti-bacterial activities (Ma, 2014). Anthocyanins are the major constituents of N. tangutorum. Bai (2009) studied the effects of purified anthocyanins from N. tangutorum fruits on lipid levels and lipid peroxides of hyperlipidemic rats, which showed that N. tangutorum fruits have significantly lower blood fat and anti-oxidant effect. The study on the antioxidant activity of the varieties of N. tangutorum by Zheng et al. (2011) further demonstrated that the antioxidant activity of N. tangutorum fruits was significantly correlated with the anthocyanin content. The animal experiment conducted by Ma et al. (2016) showed that anthocyanin components have potential biological activity both in vivo and in vitro. Li et al. (2019) explored the effect of anthocyanins from N. tangutorum fruits on non-alcoholic fatty liver injury (NAFLD) induced by high-fat diet (HFD) in mice and its underlying mechanism. The result showed that anthocyanins from N. tangutorum fruits can ameliorate HFD-induced NAFLD in mice by regulating oxidative stress and lipid metabolism in hepatocytes. Zhang et al. (2017) and Wu et al. (2015) studied anthocyanins obtained by fractionation from industrial by-products of N. tangutorum juice and showed that anthocyanins in N. tangutorum also play an important role in cardioprotective effects. Jiang et al. (1994) analyzed aging-related enzymes and metabolites of mice after feeding with N. tangutorum juice and showed that N. tangutorum juice has certain anti-aging, nutrition, and health benefits. Suo & Wang (2004) fed hyperglycemic and normal mouse models with N. tangutorum juice and measured the blood glucose levels of the experimental animals. The results showed that the fruits of N. tangutorum had a good hypoglycemic action. Liu et al. (2023) revealed through in vitro and in vivo experiments that the pharmacological basis of the hypoglycemic action of N. tangutorum fruits may be diphenylpropanoids. In addition, the experimental results of Li Bing also suggest that N. tangutorum fruits can improve cognitive dysfunction and protect hippocampal neurons in type 2 diabetic rats (Li, 2020b). Polysaccharides are one of the chemical components of N. tangutorum. Meng et al. extracted and purified water-soluble polysaccharides from the fruit of N. tangutorum, and showed that the polysaccharides have anti-inflammatory and anti-oxidant biological activities by studying their effects on lipopolysaccharide-induced acute lung injury in mice (Meng et al., 2021). Zhao et al. (2018) verified the results of their study by conducting experiments on the purification of polysaccharides from N. tangutorum fruits and the evaluation of free radical scavenging ability in vitro. The seed oil of N. tangutorum contains abundant health-promoting active ingredients, and is a rare functional oil rich in unsaturated fatty acids in nature. Suo, Gao & Wang (2005) studied the security and blood lipid lowering effect of N. tangutorum seed oil from Qinghai Qaidam Basin and found that N. tangutorum seed oil can remove cholesterol from blood vessel wall, with a prominent blood lipid lowering effect. Liu et al. (2014) determined by response surface methodology (RSM) that the optimal extraction parameters for subcritical fluid extraction (SFE) of N. tangutorum seed oil were an extraction time of 40 min, an extraction pressure of 0.60 MPa, an extraction temperature of 44 °C, and a raw material particle size of 0.45 mm.

Economic value

The economic value of N. tangutorum is mainly reflected in windbreaks and sand fixation, animal husbandry, beverage production, food additives and medicine. N. tangutorum is drought-tolerant, salt-loving, multi-branched, and sand-resistant. It is easy to form new branches and then form N. tangutorum nebkha after sand burial, which can hold and fix a lot of quicksand (Wang et al., 2007; Ding, 2020). According to the study of Guo et al. (1993) N. tangutorum nebkha with 2 m height and crown width of 11.3 m × 10.3 m can accumulate up to 2,231.3 m3 of sand. The branches and leaves of N. tangutorum have high nutritional value, rich in amino acids, crude protein, crude fat, soluble sugars as well as mineral elements such as phosphorus, iron, calcium, zinc, etc. They are good feed for the development of animal husbandry (Zhao, 2023). Sugar and acid are important nutrients in N. tangutorum fruits, and the sugar-acid ratio is the basic index of brewing alcoholic beverages (Li, 2007). Meng et al. (2023) analyzed the physicochemical indices, phenolic substances and aroma components of N. tangutorum fruits from five natural populations distributed in Qinghai Province. The results showed that N. tangutorum fruits have a high content of reducing sugars and phenolic substances, and a variety of volatile substances species. Therefore, the fruit of N. tangutorum has great development potential in dry red wine brewing. Wang et al. (2021) brewed dry red wine of N. tangutorum by aging and compared the nutritional components with commercial dry red wine, which showed that dry red wine of N. tangutorum has higher nutritional value. Anthocyanins are the main components of N. tangutorum dry red wine. The optimal extraction conditions for anthocyanins of N. tangutorum fruit were obtained with response surface methodology (RSM) by Li Bing, and the conditions were 70% methanol extraction for 32 min under 70 °C (Li, 2020a). Natural food coloring is increasingly popular of people due to its safety, reliability, natural color, nutritional and pharmacological effects. Hu et al. (2014) isolated four types of anthocyanins from N. tangutorum fruits by preparative HPLC. And they considered that these anthocyanins are ideal natural pigment resources because of the high stability and anti-oxidant activity. Meng et al. (2006) in their study on the stability of red pigment from N. tangutorum fruits showed that the red pigment in N. tangutorum fruits has good stability and non-toxicity. It can be widely used as a food additive in food, health products and medical industry. N. tangutorum has high medicinal value and is used in folk medicine to treat spleen and stomach weakness, indigestion, neurasthenia and other diseases (Yang et al., 2013b). Meanwhile, the root of N. tangutorum is the specific host of Cynomorium songaricum, which is a kind of traditional herbal medicine. It has ideal effects on enhancing immunity, anti-aging, regulating endocrine and treating age-related diseases (Gao, 1996). Wang (2012) directly inoculated the germinated Cynomorium seed in the root of N. tangutorum after breaking their skins, which could quickly establish a parasitic relationship with the three-year-old transplanted seedlings and obtain economic benefits.

PROSPECTS

N. tangutorum is a perennial shrub of the Nitrariaceae family with high stress resistance. It is an excellent tree for windbreak and sand fixation in the Gobi and desert areas. In recent years, the species has suffered varying degrees of degradation as a result of excessive deforestation, overgrazing as well as the deterioration of natural conditions in desert areas. Therefore, the cultivation and screening of N. tangutorum plants with strong stress resistance is of great significance for the management of deserts, the reformation of saline-alkali land, the improvement of the ecological environment as well as the improvement of the living conditions of local residents. At present, research on the stress resistance of N. tangutorum is mainly focused on evaluating of stress resistance of different families and the physiological and biochemical changes under stress. And there are few studies on the stress resistance of N. tangutorum at the molecular level. In the follow-up research, we should focus on using molecular means to explore the metabolic pathways and key genes under adversity stress, so as to lay a theoretical foundation for the cultivation of stress resistant plants of N. tangutorum. To explore the genetic transformation system of N. tangutorum and cultivate N. tangutorum plants with strong stress resistance. As the research on N. tangutorum deepens, its application in fruit wine, fruit oil, cosmetics and health products are becoming increasingly widespread. However, compared with characteristic resources such as Lycium, the research foundation of edible and pharmacological research on N. tangutorum is still weak, and the degree of product development is limited. Scholars at home and abroad have conducted extensive research on the chemical composition and pharmacological effects of N. tangutorum and have made some progress. However, previous research on the chemical constituents of N. tangutorum mainly focused on flavonoids, anthocyanins, polyphenols, alkaloids, and polysaccharides, with less involvement in other types of studies. Medical effectiveness studies have mainly focused on the efficacy of anti-inflammatory, anti-oxidant and treatment of various major tumors, and there are few studies on the efficacy of other diseases. Moreover, the pharmacological activity test still remains at the level of the crude substance, and the correlation study between monomeric compounds and pharmacological activity is inadequate. The extraction technology of monomeric compounds is less studied and the amount of extraction is very limited. Therefore, further studies should enhance the development of chemical constituents of N. tangutorum and expand its industrial development scope. To study the efficacy of the chemical constituents of N. tangutorum and broaden the medicinal value of N. tangutorum. In-depth research on the pharmacological effects of monomeric compounds to promote the development of precision medicine in N. tangutorum. Intensify research on the extraction technology of N. tangutorum to increase the amount extracted. N. tangutorum has a wide distribution, mainly in the northwestern arid zone at altitudes between 2,100 and 3,100 meters. The distribution of N. tangutorum in the Qaidam Basin is particularly widespread, with a total area of about 5.0 ×105 acres, an average annual yield of more than 70 kg per acre, and an annual fresh fruit production of more than 3.5 × 104 tons. This provides a solid raw material foundation for the development of the N. tangutorum industry. N. tangutorum has great potential for exploitation and utilization. However, excessive development and utilization will break the balance of the ecological environment, destroy the quality of the ecological environment and deteriorate the fragile ecosystem. Therefore, the development and utilization of N. tangutorum should be combined with the concept of ecological environmental governance, adhere to the principles of protection priority, scientific planning, and rational utilization, and realize the organic unity of economic construction and ecological governance.

Additional Information and Declarations

Competing Interests

Author Contributions

Data Availability

The authors declare there are no competing interests.

Xiaolan Li conceived and designed the experiments, analyzed the data, authored or reviewed drafts of the article, and approved the final draft.

Hanghang Liu performed the experiments, analyzed the data, authored or reviewed drafts of the article, and approved the final draft.

Chaoqun Li performed the experiments, authored or reviewed drafts of the article, and approved the final draft.

Yi Li conceived and designed the experiments, analyzed the data, authored or reviewed drafts of the article, and approved the final draft.

The following information was supplied regarding data availability:

This is a literature review.

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
