# Peer review of "A systematic review on the morphology structure, propagation characteristics, resistance physiology and exploitation and utilization of Nitraria tangutorum Bobrov"

_PeerJ, doi:10.7717/peerj.17830_

## Round 0.1 · original submission · Major Revisions

The review needs substantial revision. Revise the review as per the following suggestions and the reviewer's comments.

Major comments:

1. Change the title- In the current form it looks very general and does not give much information about the content and/or importance of the studied plant.

2. Revise the abstract. It must be more informative and not just a summary of the whole review.

3. Add a figure/s depicting the salt tolerance mechanism and/or utilization of the studied plant.

4. Add a comparative table showing the potential of the studied plant and other halophyte/Xerophytic plants in terms of tolerance level and economic value.

5. You may go through the following review—“Sesuvium portulacastrum, a plant for drought, salt stress, sand fixation, food, and phytoremediation. A review”

6. In the acknowledgment section, no need to acknowledge the reviewer.

·

Basic reporting

I find the language to be acceptable with some necessary clarification of several terms requested in General Comments. The disciplinary focus of the paper is a bit unclear, though of interest to me as a potential plant from nature that could have economic purpose. I hope there will also be further emphasis on the ecological value also.

Experimental design

There was relatively minimal information about the study design provided. Authors selected several papers for review detailed in the reference list. Were each of these papers cited in the article text? What was the overall composition and context of those papers? A summary seems to lack. Instead, the review seems to follow in a loose pattern of "here is a general statement about things we would like to know about or do with plants generally" followed by a "here's a paper about that with N. tangutorum". What do we learn specifically about N. tangutorum and what is knowledge that can be generalized more broadly?

Validity of the findings

My overall sentiment is that the review outcome and commentary has many areas for potential improvement. It is clear there is an abundance of information about the plant with interesting commentary, though I am not sure if a prior review was completed. Basic description of the plant, it's physiology, and ecology are not well organized, but scattered throughout the review. Many sections seem out of place or poorly organized related to overarching findings and summary of knowledge related to the plant. Instead, an abundance of examples are combined to form paragraphs.

Additional comments

My hope is to provide a constructive review about this article related to a plant for which I am not familiar. I am an expert in agroecology and the integration of plants occurring in nature and agriculture. I list detailed line comments to express in detail the comments made in the Basic Reporting, Study Design and Validity of Findings structure. In my review, I refer to N. tangutorum as "the plant".

8: xerophytic and halophytic should be defined at first use in the abstract. Well described in Line 368.
9: in what ways is the plant constructive?
16: if N. tangutorum is a constructive part of the desert ecosystem, specific detail should be included on how to avoid exploitation from the sense to not compromise the native population by excessive extraction and associated ecological harm

30: the plant is stated to be common, so then in what way could the fruit be "rare"?
39: The paragraph is a bit confusing to the reader in relation to the description of the whole-plant characteristics and also in relation to each of the plant parts. For example, this description of seed oil qualities follows references on heath effects associated with "it". Clarify whole plant or plant part for the "it".
46: What is it that has limited the use and economic potential thus far? Has there been commercial attempt previously?

49: Be specific how many articles resulted from the search that "appeared in the database"

54: I believe morphological description has a standard form. Or at least I expect details organized for a botanical description: growth habit, leaves, flowers and fruit. Once that is established the additional commentary can follow. Also, please provide a photo or two to support the description. You may benefit from the addition of this manuscript in your review, especially pertaining to morphology. https://peerj.com/articles/14934/

57: "This plant is low and small [with] clump, middle branches, [and] ..."
63: not sure what is meant by "which playing great significant role for the identification of plant species and varieties"
73: What is meant by "decorative type"?
78: I am concerned by the presumption of "exploitation" related to ecological impact.

85: This section seems out of place. Complete the sections about the plant's characteristics prior to description of cultivation.
86: Separate natural dispersal and regeneration behavior for the plant from cultivation/scientific/agricultural methods.
102: variation in morphology was described as a valuable quality for the plant, but now it is stated "stability of excellent properties" is preferred. Comment further to the use case and relevant choice.
108: Has there been regeneration demonstrated after callus formation? Why use tissue culture when readily roots from cuttings? What about propagation from seed?

118: Describe as overview the relative status of the plant's drought resistance and summarize the features of the plant that contribute to that characteristic. Like Line 169. It is also not clear to me how subsection Effects (119) differs from subsection Response (152).

163: Family is a very specific botanical term. I believe Nitraria is in the family Nitrariaceae. What then is meant here by "family"?

187: I also suggest a general statement about salt tolerance here. Like Line 215. It seems like the plant was used as a model for various physiological studies. Perhaps these efforts could be summarized also followed by the description of findings. Are any of these characteristics determined to be unique to the plant or are they model observations likely generalizable to similar (or all) plants?

287: What is the goal(s) of exploitation and utilization? What have been the barriers thus far? What might explain the discrepancy between research output and harvest from natural or agricultural settings? What plant parts are harvested? What is a reasonable harvest? Would a whole population be harvested? I would expect some caution on how to collect from nature but preserve the population while also consider populations under cultivation. Some relevant information appears at Line 438.

288: I think this chemical composition section is out of place and may follow as characteristics of morphology. Is there evidence of this plant being used for its chemicals?

366: I expect economic value to reflect harvest rate and market value. Here is presented more like use or harvest potential or "Prospects".

404: It may be this mention of Zygophyllaceae is out of date or synonymous with the Nitrariaceae. Please, provide a botanical reference to confirm.

406: This review should be very careful to avoid further degradation by "exploitation". Also, is desert restoration a viable opportunity for land managers to include and increase the plant's populations.

414: The molecular goals of this paper are apparent throughout, but were not presented clearly at the outset. I had ecological protection and agricultural use in mind, whereas the authors seem to present a molecular model plant instead. Is there a clear priority? Perhaps this is a reflection on the reviewed science, but misses the mark for me on the set up from the abstract.

Reviewer 2 ·

Basic reporting

Clarity and Structure: Assess whether the paper has a clear structure with a logical flow of information. Ensure that the introduction provides adequate context for Nitraria tangutorum, including its significance in desert ecosystems.
Language and Grammar: Check for grammatical errors and awkward phrasing. Suggest improvements where necessary for clarity and readability.
Use of Figures and Tables: Review the quality and relevance of figures and tables. Are they properly labeled and do they complement the text? Recommend any adjustments for improved clarity, such as adding legends or adjusting scales.
Literature Review and References: Examine the thoroughness of the literature review. Does the paper provide a comprehensive background on Nitraria tangutorum, including its ecological role, distribution, and any previous research? Verify the accuracy of references and suggest additional relevant studies if needed.

Experimental design

Research Question and Hypothesis: Determine if the study's research question is clearly defined and if the hypothesis is explicitly stated. Assess whether the hypothesis is testable and aligns with the overall research goal.
Methodology and Data Collection: Review the study's methodology for robustness and reproducibility. Does the paper provide sufficient detail for other researchers to replicate the study? Comment on the sampling methods, experimental design, and data collection techniques. If the study involves fieldwork, assess whether the locations and conditions are well-documented.
Statistical Analysis: Evaluate the statistical methods used. Are they appropriate for the type of data collected? Recommend additional analyses if the current ones seem insufficient or inappropriate. Ensure that the paper reports significant results correctly and uses appropriate statistical tests.

Validity of the findings

Data Interpretation and Conclusions: Examine the interpretation of the results. Do the conclusions logically follow from the data? Suggest improvements if the analysis appears biased or the conclusions are overstated.
Consistency and Reliability: Assess the consistency of the findings across the data set. Are there any anomalies or outliers that need further explanation? Comment on the reliability of the data and results.
External Validity and Generalizability: Consider whether the findings are generalizable to broader contexts or if they are specific to the study's conditions. Discuss the potential impact of environmental or ecological variables on the results.
Limitations and Bias: Look for any limitations in the study design or methodology that could impact the validity of the findings. Suggest ways to address these limitations or acknowledge their presence. Identify any potential sources of bias and recommend corrective measures.

Additional comments

Ethical Considerations: Ensure the study adheres to ethical guidelines, particularly if it involves human or animal subjects. If the study involves fieldwork, confirm that appropriate permissions and permits were obtained.
Relevance and Contribution: Consider the relevance of the study to the broader field of desert plant research and its potential contribution to knowledge about Nitraria tangutorum. Suggest additional areas of focus if applicable.

---

## Round 0.2 · accepted · Accept

Although the prior reviewers were not available to re-review, I can confirm that the authors have addressed all the comments and it is ready for publication.